# FOF: Learning Fourier Occupancy Field for Monocular Real-time Human Reconstruction

**Qiao Feng**
Tianjin University
fengqiao@tju.edu.cn

**Yebin Liu**
Tsinghua University
liuyebin@mail.tsinghua.edu.cn

**Yu-Kun Lai**
Cardiff University
laiy4@cardiff.ac.uk

**Jingyu Yang**
Tianjin University
yjy@tju.edu.cn

**Kun Li**[*]
Tianjin University
lik@tju.edu.cn

## Abstract

The advent of deep learning has led to significant progress in monocular human reconstruction. However, existing representations, such as parametric models, voxel grids, meshes and implicit neural representations, have difficulties achieving high-quality results and real-time speed at the same time. In this paper, we propose *Fourier Occupancy Field (FOF)*, a novel, powerful, efficient and flexible 3D geometry representation, for monocular real-time and accurate human reconstruction. A FOF represents a 3D object with a 2D field orthogonal to the view direction where at each 2D position the occupancy field of the object along the view direction is compactly represented with the first few terms of Fourier series, which retains the topology and neighborhood relation in the 2D domain. A FOF can be stored as a multi-channel image, which is compatible with 2D convolutional neural networks and can bridge the gap between 3D geometries and 2D images. A FOF is very flexible and extensible, *e.g.*, parametric models can be easily integrated into a FOF as a prior to generate more robust results. Meshes and our FOF can be easily inter-converted. Based on FOF, we design the first 30+FPS high-fidelity real-time monocular human reconstruction framework. We demonstrate the potential of FOF on both public datasets and real captured data. The code is available for research purposes at *http://cic.tju.edu.cn/faculty/likun/projects/FOF*.

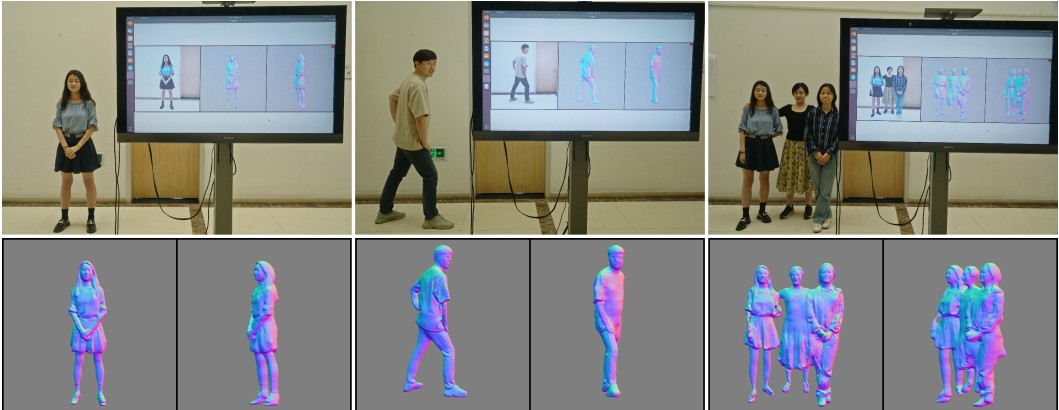

Figure 1: Our novel 3D geometry representation (FOF) enables monocular real-time human reconstruction with high-quality results at 30+FPS.

---

[*]Corresponding author

36th Conference on Neural Information Processing Systems (NeurIPS 2022).

Table 1: Comparison with existing representations. (Generality: ability to deal with complex topologies and geometries.)

| Representation | Aligned with Input | High-Quality | Computational Efficiency | Generality |
|---|---|---|---|---|
| Parametric Model [17, 20] | ✗ | ✗ | ✓ | ✗ |
| Voxel Grid [31] | ✓ | ✗ | ✗ | ✓ |
| Mesh [33, 2, 14] | ✗ | ✓ | ✓ | ✗ |
| Implicit Function [22, 23, 30, 9, 3, 27] | ✓ | ✓ | ✗ | ✓ |
| **Our FOF** | ✓ | ✓ | ✓ | ✓ |

# 1  Introduction

3D human reconstruction from a single image is very popular in computer vision and graphics, which can be widely used in various applications, such as mixed reality, virtual try-on, and "metaverse". However, high-fidelity monocular human reconstruction in real-time remains a challenging task. 3D representation is at the core of this problem and determines the design and performance of approach. 3D geometry representation is also the key for 3DTV and Holographic Telepresence systems, which need to fulfill the requirements of accuracy, efficiency, compatibility and generality [1].

Parametric human models, *e.g.*, SMPL [17] and SMPL-X [20], represent a 3D human body with parameters. Their results are robust but they are unable to reconstruct the clothes. Many classic representations, such as voxel grids [31] or meshes [33, 2, 14], have been exploited for monocular human reconstruction. However, the representation of voxel grids has to store discrete spatial grid samplings in memory. It has a space complexity of $O(n^3)$ (where $n$ is the number of grids in each dimension), which limits the achievable spatial resolution. Meshes on the other hand are a compact representation for surface geometry. However, it is challenging for meshes to cope with change of connectivity, and as a result, they may produce unstable or over-smoothed results when there are change of topology and/or substantial deformations.

Recently, implicit neural representations have emerged and been widely used in monocular human reconstruction [22, 23, 30, 9, 3, 27]. The 3D space is treated as a continuous field, such as an occupancy field and signed distance field, which can be formulated as $\mathbb{R}^3 \to \mathbb{R} : F(x, y, z)$. Rather than sampling the field and store it as a voxel grid, a neural network is used to fit the field. Thus, the results can be produced at any resolution. However, human geometry is not explicitly reconstructed, and the network has to infer values for a huge number of spatial grid sample points to extract the geometry. The number of sampled points in inference is limited by available GPU memory, and the sampled points have to be forwarded in several batches, which is very time-consuming. Therefore, it is very difficult to implement high frame rate real-time reconstruction based on implicit neural representations. Although Monoport [15], as a real-time method, renders the mesh-free implicit field at 15FPS with an efficient sampling scheme, it does not reconstruct the geometry explicitly. Thus, it is not compatible with traditional graphics applications.

In this paper, we propose a new representation, *Fourier Occupancy Field (FOF)*, for monocular real-time and accurate human reconstruction, which is an expressive, efficient and flexible 3D object representation in the form of a 2D map aligned with the input image. The key idea of FOF is to represent a 3D object with a 2D field by representing the occupancy field along the $z$-axis into Fourier series and only keeping the first few terms, which is memory-efficient and can preserve most of the geometry information. With grid sampling on the $x$-axis and $y$-axis, the FOF can be stored explicitly as a multi-channel image, which is compatible with the existing CNN frameworks and suitable for 3DTV applications. Unlike depth maps, our FOF can represent the complete geometry rather than just the visible part, and it is very efficient to produce high-fidelity human shapes. Moreover, high-resolution meshes and our FOF can be easily inter-converted. Detailed comparisons with existing representations are given in Table 1.

Based on our FOF, we design a simple and efficient pipeline for monocular real-time human reconstruction, which is end-to-end and easy to train. Some live captured examples are shown in Fig. 1. Experimental results on both public dataset and real captured data show that our approach can reconstruct human meshes accurately and robustly in real-time.

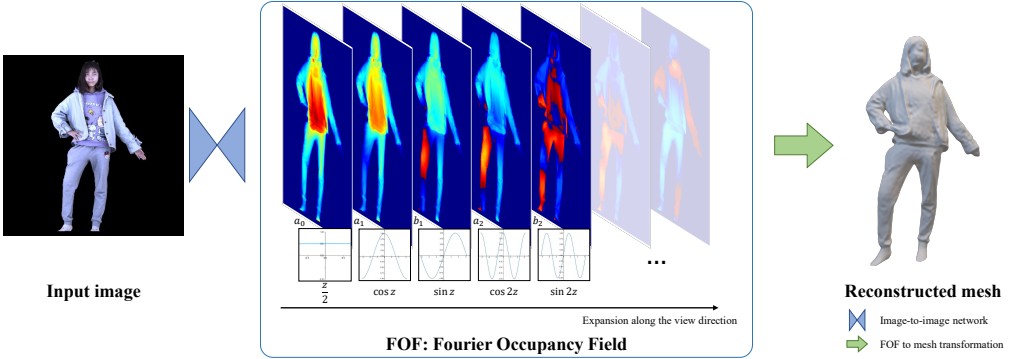

Figure 2: Overview of our reconstruction pipeline. With our FOF, only a image-to-image network is needed for high quality monocular human reconstruction.

To summarize, the main contributions of our work include:

- We propose *Fourier Occupancy Field*, a novel representation for 3D humans, which can represent a high-quality object geometry with a 2D map aligned with the image. It can bridge the gap between 2D images and 3D geometries.

- We present a simple and efficient pipeline for monocular real-time 3D human reconstruction, which is end-to-end and easy to train. Parametric models can be optionally transformed into a Fourier occupancy field as a prior to produce more robust results.

- Compared with SOTA methods based on other representations, our approach can produce more realistic results in real-time. Moreover, it is the first 30+FPS pipeline, which is 2 times faster than Monoport [15] with better results.

## 2  Related work

We briefly overview literature on monocular human reconstruction using a single color camera here.

**Surface-based reconstruction.**    This kind of approaches focuses on the interface between geometry and empty space and can be broadly classified into three categories: parametric models, UV-map-based methods, and graph-based methods. Parametric models, such as SMPL [17], SMPL-X [20] and SRAR [19] are popular representations for 3D human. They collect a large amount of naked human shapes, and get an analytical model on them with statistical methods, which can generate a human body mesh with dozens of parameters. Based on this, a lot of works [12, 29, 13] estimate 3D human shapes from an RGB image by predicting the parameters of the parametric model, but these methods cannot reconstruct the clothes or hair. An overview of 3D human shape estimation from a single image can be found in Tian *et al*. [25]. Other methods try to estimate clothed humans by adjusting vertices of the parametric models. Alldieck *et al*. [2] warp the input image to align with the UV map, and estimate the displacements on the T-pose SMPL mesh. Their results are always in T-pose and cannot match the input images well. Li *et al*. [14] reconstruct the geometry of a clothed human body with graph neural networks, but the recovered mesh tends to be smooth and lacks fine details. Zhu *et al*. [32, 33] use four stages constrained by joints, silhouettes and shading, but their results are sometimes inconsistent with the images. It is very difficult to represent 2D manifold surfaces of different 3D objects with a fixed structure, and hence most methods above rely heavily on the topology defined by SMPL and cannot produce detailed human geometries with complex topology. Habermann *et al*. [7, 8] propose a deep learning approach for monocular dense human performance capture, but this method needs to scan the actor with a 3D scanner as the template mesh. Graph-based methods cannot get exact features aligned with the results, and hence the results are unstable and over-smoothed.

**Volume-based reconstruction.**    Volume-based methods estimate the attributes, *e.g.*, occupancy and signed distance, for each point in the 3D space and can represent 3D shapes with arbitrary topology. Voxel girds and implicit neural networks are two main representations used in volume-based methods.

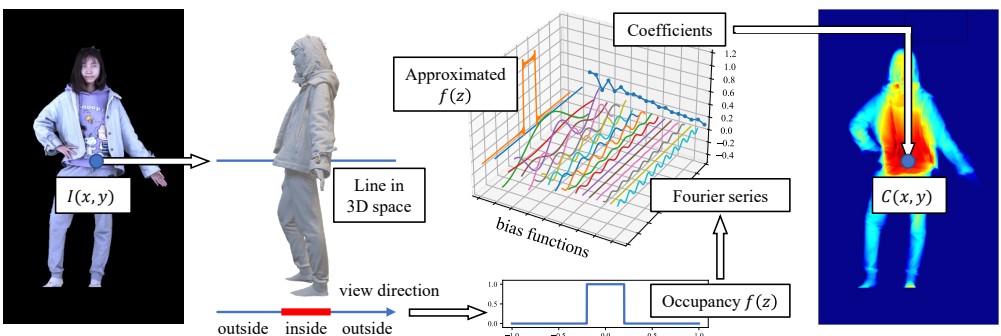

Figure 3: The proposed Fourier Occupancy Field (FOF). Each pixel on the image $I$ corresponds to a line in 3D space and can be described with a occupancy function $f(z)$, which can be expanded as Fourier series and represented with a vector $C(x, y)$. The 3D human model can be encoded as a 2D vector field, our FOF.

Zheng *et al.* [31] regress the voxel grids of 3D human using convolutional neural networks, but this kind of methods requires intensive memory and has limited resolution of the model. Unlike voxel grids, implicit neural representation can represent detailed 3D shapes without resolution limitations. Saito *et al.* [22] use pixel-aligned function to reconstruct 3D human from a single RGB image. They further use normal maps to significantly improve the 3D geometric details [23]. Zheng *et al.* [30] use the voxelized SMPL mesh as a prior to make the results more robust. Xiu *et al.* [27] propose a method to correct SMPL model estimated by other methods [29, 13, 6] according to the input image, and then regress occupancy field from the normal map and the signed distance field of the corrected SMPL. Their results are robust to human poses, but the geometric details do not match the input images and are noisy. The common problem of the methods based on implicit neural representation is that they are computationally heavy and very time-consuming. Although Li *et al.* [15] propose an efficient sampling scheme to speed up the inference, they can only achieve mesh-free rendering at 15FPS. When generating a complete mesh, their method will be less than 10FPS.

In this paper, we contribute the first real-time high-quality human reconstruction system working at 30+FPS with a single RGB camera, benefiting from the proposed efficient and flexible 3D geometry representation FOF.

## 3 Method

The goal of our work is to reconstruct a high-fidelity 3D human model from a single RGB image in real-time. To this end, we propose an efficient and flexible 2D representation of 3D geometry (Sec. 3.1), named *Fourier Occupancy Field (FOF)*, as shown in Fig. 2. In the proposed FOF-based reconstruction framework, we first learn the FOF representation for the input image by an image-to-FOF mapping (Sec. 3.2), and then reconstruct the 3D human model by the efficient Fourier inversion with low complexity (Sec. 3.3).

### 3.1 Fourier Occupancy Field

Without loss of generality, 3D objects to be reconstructed are normalized into a $[-1, 1]^3$ cube. Denoted by $(x, y, z) \in [-1, 1]^3$ the 3D coordinates, and by $S$ the surface of a 3D object. The 3D object can be described by a 3D occupancy field $F : [-1, 1]^3 \mapsto \{0, 0.5, 1\}$, where the occupancy function $F(x, y, z)$ is defined as

$$F(x, y, z) = \begin{cases} 1, & (x, y, z) \text{ is inside the object,} \\ 0.5, & (x, y, z) \in S, \\ 0, & (x, y, z) \text{ is outside the object.} \end{cases} \tag{1}$$

Such a 3D representation is highly redundant, because only a small subset, *i.e.*, the iso-surface of $F(x, y, z)$ with value of 0.5, is sufficient to represent the surface of the object. Direct inference of the 3D occupancy field $F$ from a single image not only confronts with the curse of dimensionality, but also requires more computation and memory in handling high dimensional feature maps. Note

that the occupancy field defined in the 3D cube can be regarded as the collection of a large number of 1D signals defined on the lines along the view direction. Without loss of generality, we assume that the view direction is the same as the $z$-axis of the 3D cube. The occupancy function, as a 1D signal, along the line passing through a particular point $(x^*, y^*) \in [-1, 1]^2$ on the $xy$-plane can be written as $f(z) : [-1, 1] \mapsto F(x^*, y^*, z)$. We explore Fourier representation for such a family of 1D occupancy signals, and propose a more compact 2D vector field orthogonal to the view direction, namely FOF, for more efficient representation of the 3D occupancy field, which is detailed in the following subsections.

**Fourier series on a single occupancy line.** 1D occupancy signals $\{f(z)\}$ are essentially on-off signals switching at the boundaries of human bodies, which lie in a low-dimensional manifold in the ambient signal space. For accurate and fast 3D reconstruction, the compact representation should have the following merits:

1. *Sampling Scalability*: The representation should be able to adapt to different sampling rates in the inference stage without introducing systematic mismatch besides the inherent approximation error due to sampling. Without retraining, systems with such a representation would support diverse applications with different resolution, quality, and speed requirements.

2. *Stable Reconstruction*: The representation should provide stable reconstruction quality for diverse humans with various parameters such as garments, heights, weight, and poses.

3. *Low-complexity Reconstruction*: The representation should allow low-complexity reconstruction to support emerging real-time applications such as Holographic Telepresence.

However, most popular representation do not enjoy all the above merits. For instance, discrete Fourier transform and discrete wavelet transform have stable and low-complexity reconstruction, but only support the sampling rate used in preparation of the training set due to their discrete nature. Learned dictionaries are able to provide sparse representation, but require computation-demanding iterative reconstruction, leaving alone the sampling scalability. Without doubt, sampling scalability is the most difficult property to meet. Pushing the concept of sampling scalability to the limit, one representation would be scalable to arbitrary sampling rates in theory if it allows continuous reconstruction along the $z$-axis. Note that the representation should stay discrete and compact for efficient inference. Such a bridge of discrete-representation and continuous-reconstruction motivates us to use Fourier series for the representation of 1D occupancy signals $\{f(z)\}$. By nature, Fourier series would have infinite number of discrete coefficients, which is impractical. Note that most energy of 1D occupancy signals concentrate on only a few low-frequency terms, which yields a compact representation by subspace approximation.

Formally, let $f_p(z)$ be periodic extension of $f(z)$. Note that $f_p(z)$ satisfies the Dirichlet conditions: 1) $f_p(z)$ is absolutely integrable over one period; 2) $f_p(z)$ has a finite number of discontinuities within one period; 3) $f_p(z)$ has a finite number of extreme points within one period. Thus, $f_p(z)$ can be expanded as a convergent Fourier series:

$$f_p(z) = \frac{a_0}{2} + \sum_{n=1}^{\infty} \left( a_n \cos(nz\pi) + b_n \sin(nz\pi) \right), \tag{2}$$

where $\{a_n\}, \{b_n\} \in \mathbb{R}$ are coefficients of basis functions $\{\cos(nx)\}$ and $\{\sin(nx)\}$, respectively. Since $f(z)$ is a specific period of $f_p(z)$, Eq. (2) also holds for $f(z)$. Note that by defining the occupancy function as Eq. (1), Eq. (2) holds for discontinuity at $z^*$ on the surface $S$ because $f(z^*) = (f(z^*-) + f(z^*+))/2 = 0.5$, where $f(z^*-)$ and $f(z^*+)$ are the left hand limit and right hand limit of $f(z)$ at $z^*$. For compact representation, we approximate 1D occupancy signals by a subspace spanned by the first $2N + 1$ basis functions:

$$\hat{f}(z) = \boldsymbol{b}^\top(z)\boldsymbol{c}, \tag{3}$$

where $\boldsymbol{b}(z) = [1/2, \cos(z), \sin(z), \ldots, \cos(Nz), \sin(Nz)]^\top$ is the vector of the first $2N + 1$ basis functions spanning the approximation subspace, and $\boldsymbol{c} = [a_0, a_1, b_1, \ldots, a_N, b_N]^\top$ is the coefficient vector which provide a more compact representation of the 1D occupancy function $f(z)$. In our implementation, $N$ is chosen as 15, which is accurate enough for most 3D human geometries.

**Fourier occupancy field.** Then, such a Fourier subspace approximation is applied to all 1D occupancy signals over the $xy$-plane, which is shown in Fig. 3.

$$\hat{F}(x, y, z) = \boldsymbol{b}^\top(z)\boldsymbol{C}(x, y), \tag{4}$$

where $\boldsymbol{C}(x, y)$ is the $(2N + 1)$ Fourier coefficient vectors for the 1D occupancy signals at $(x, y)$. In this way, we obtain the *Fourier Occupancy Field* $\boldsymbol{C} : [-1, 1]^2 \mapsto \mathbb{R}^{2N+1}$ for 3D occupancy field $F$.

Note that 3D coordinates $(x, y, z)$ in Eq. (4) are continuous, and can be sampled as needed. Particularly, unlike the discrete Fourier transform or discrete wavelet transform, the subspace dimension for 1D occupancy signals $\{f(z)\}$ is not coupled with the sampling grid of $z$-axis. This would avoid potential sampling mismatch between the sampling grid used in the training data and that in the testing stage. The reconstruction of $\hat{F}$ from the FOF $C$ is sampling scalable in the sense that sampling rate along the $z$-axis can be chosen as necessary, *e.g.*, according to the requirement of reconstruction quality and speed. 3D models at any resolution can be readily obtained from the FOF representation by simply alter the sampling rate along the $z$-axis. Moreover, with sampling on $xy$-plane, *e.g.*, aligned with the sampling grid of the input image of size $H \times W$, the FOF $C$ is a multi-channel image of size $H \times W \times (2N + 1)$, which is more compact than the 3D occupancy field and is friendly for network prediction as detailed in the following subsection.

## 3.2 Learning FOF with neural networks and variants

**Network.** Thanks to the efficient dimensionality reduction with subspace approximation, the task of FOF learning from an input image is essentially to learn an image-to-image network. To this end, we exploit the network standing on the HR-Net [26] framework for its outstanding fitting capability in various vision tasks. We use the weak perspective camera model in our implementation, in which FOF (and thus the reconstructed geometry) is naturally aligned with the input RGB image. Note that perspective camera model can also be used by using the normalized device coordinate space (NDC space). Note that the proposed FOF representation is scalable in terms of sampling along not only the $z$-axis (discussed in Sec. 3.1), but also the $x$- and $y$-axes. For example, for the input image of size $W \times H$, we learn the FOF of size $W \times H \times (2N + 1)$. To reconstruct a 3D shape of size $W' \times H' \times K$, we simply resize the learned FOF to the size of $W' \times H' \times (2N+1)$ along the $xy$-plane, and specify the number of sampling points $K$ for the $z$-axis at the Fourier inversion. Such a scalable reconstruction of 3D shapes is a significant departure from scaling-after-reconstruction strategies that involve complex 3D manipulation, and is more analogous to efficient image resizing. Without requiring complex accommodation, our method can be readily deployed in various applications with heterogeneous computation resources. Our FOF is also highly extensible. Here we introduce two variants of the reconstruction pipeline, named FOF-SMPL and FOF-Normal.

**FOF-SMPL.** In FOF-SMPL, we use the SMPL parametric model as a prior to achieve more robust reconstruction, inspired by PaMIR [30]. Specifically, we transform a SMPL mesh to a FOF with the algorithm in Sec. 3.3, and concatenate it with the RGB image as the input together. Thus, the input of FOF-SMPL is a $(31 + 3)$-channel image. The exploit of smpl provides a reference for the result, and eliminates the depth ambiguity which makes the network more focused on the local relative details rather than the global absolute 3D positions. This makes the network more robust to different poses and improves reconstruction quality for cases with high ambiguity.

**FOF-Normal.** In FOF-Normal, we use normal maps to enhance the results. Specifically, we infer the front and back normal maps with the pix2pix network [11] for the input image, and also concatenate them with the RGB image together as input. Thus, the input of FOF-Normal is a $(3 + 3 + 3)$-channel image. Note that we use the inferred normal maps, instead of ground-truth, in the training stage, which makes the network more robust for the input normal maps.

**Loss.** We use the L1 loss is to train our FOF baseline and variants. To make the network more focused on the human geometry, we only supervise the human foreground region of the image. Given a training sample $(I, C)$, the loss function $\mathcal{L}$ is formulated as:

$$\mathcal{L} = \frac{1}{|\mathcal{M}|} \sum_{(x,y)\in\mathcal{M}} \left\| \hat{\boldsymbol{C}}(x, y) - \boldsymbol{C}(x, y) \right\|_1, \tag{5}$$

where $\mathcal{M}$ is the set of foreground pixels of $C$, and $\hat{C}$ is the FOF predicted by the network.

## 3.3 Transformation between FOF and mesh

**FOF to mesh.** Extracting meshes is necessary for applications such as mesh-based rendering and animation. Recent methods based on implicit neural representations need to run a network on a 3D sampling grid. Instead, as described in Eq. (4), the reconstruction of 3D occupancy field from FOF is simply a multiplication of two tensors. Then, the 3D mesh is extracted from the iso-surface of the 3D occupancy at the threshold of 0.5 with the Marching Cubes algorithm [18]. Both steps can be parallelized on GPUs for fast mesh generation. In this paper, all our results are produced directly by the marching cubes algorithm and no post-processing technique is used.

**Mesh to FOF.** To prepare training data, 3D meshes should be transformed into FOF as training labels. By definition Eq. (6), we are to calculate the first $2N + 1$ coefficients of the Fourier series for each 1D occupancy signal $f(z)$. For generic signals, this would require numerical integration over a sampled version of $f(z)$, which, however, may introduce numerical errors and demands considerable computation. Fortunately, thanks to the particular form of occupancy signals, the Fourier coefficients can be derived analytically with exact solutions. Note that the 1D occupancy signal $f(z)$ is actually defined by discontinuities, where the line goes inside or outside the mesh. We extract the discontinuous points of $f(z)$ by designing a rasterizer-like procedure. Formally, suppose the line associated with $f(z)$ passes though the mesh for $k$ times. For the $i^{\text{th}}$ traversing, let $z_i$ and $z_i'$ be the locations going inside and outside the mesh, respectively. Then, the set $\mathcal{Z} \triangleq \{(z_i, z_i')\}_{i=1}^k$ collects those intervals of the line that are inside the mesh. According to the definition of the occupancy signal, we have: $f(z) = 1, z \in \mathcal{Z}$, and $f(z) = 0$ otherwise except for a finite number of discontinuities with measure zero. Therefore, we have the following analytical solution for the Fourier coefficients.

$$\begin{cases} a_n &= \int_{-1}^{1} f(z) \cos(n\pi z) dz &= \sum_{i=1}^{k} (\sin(n\pi z_i') - \sin(n\pi z_i))/(n\pi), \\ b_n &= \int_{-1}^{1} f(z) \sin(n\pi z) dz &= \sum_{i=1}^{k} (\cos(n\pi z_i) - \cos(n\pi z_i'))/(n\pi). \end{cases} \quad (6)$$

The FOF representation is obtained by calculating the Fourier coefficients for all the 1D occupancy signals. Since the number of intervals $k$ is generally small, *e.g.*, 2∼6 for human meshes, the calculation of $a_n$ or $b_n$ is of $O(1)$ complexity. Thus, the transformation from discontinuities to FOF has a computational complexity of $O(NHW)$, and can be parallelized on GPUs for acceleration.

## 4 Experiments

### 4.1 Datasets and metrics

**Datasets.** We collect 2038 high-quality human scans from Twindom [1] and THuman2.0 [28] to train and evaluate our method. Twindom contains 1512 scans with various clothing, but their poses are simple and lack diversity. THuman2.0 contains 526 scans with more variety in body poses. We randomly select 1059 from Twindom and 368 from THuman2.0 as the training set, and 302 from Twindom and 105 from THuman2.0 as the test set. The remaining subjects are used as the validation set. We render 256 views images with 4 lights randomly selected from 240 lights for each scan. All the images are rendered with a weak perspective camera and pre-computed radiance transfer (PRT[24]) renderer. To obtain the corresponding SMPL model, which is used for FOF-SMPL and ICON [27], we use OpenPose [5] to detect 2D keypoints on the rendered images and generate SMPL model with a multi-view version of SMPLify-X [20].

**Metrics.** We use Chamfer distance, P2S (point-to-surface) distance, and normal image error for evaluation. Chamfer distance is defined as:

$$d_{CD}(S_{pr}, S_{gt}) = \frac{1}{S_{pr}} \sum_{x \in S_{pr}} \min_{y \in S_{gt}} \|x - y\|_2^2 + \frac{1}{S_{gt}} \sum_{x \in S_{gt}} \min_{y \in S_{pr}} \|x - y\|_2^2, \quad (7)$$

where $S_{pr}$ is a set of points on the predicted mesh surface, and $S_{gt}$ is a set of points on the ground-truth mesh surface. We average the Chamfer distances of all test samples, and report the square root of half of it. We also measure the average point-to-surface distance from the vertices on the reconstructed surface to the ground truth. To measure the visual quality, we render the reconstructed geometries as normal maps from the view direction, and compute L1 loss with the ground truth.

---

[1] https://web.twindom.com/

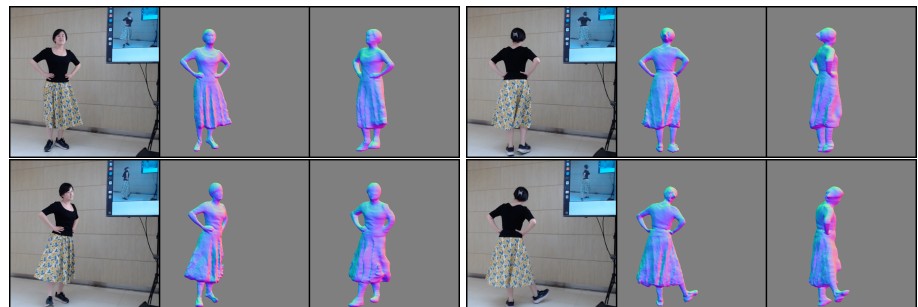

Figure 4: Real-time reconstruction results by our method.

Table 2: Comparison with the state-of-the-art methods.

|         | FOF   | FOF-SMPL | FOF-Normal | PIFu  | PIFuHD | ICON  | ICON$_{filter}$ |
|---------|-------|----------|------------|-------|--------|-------|-----------------|
| Chamfer | 2.693 | **1.580** | _2.601_   | 3.688 | 2.983  | 2.831 | 2.856           |
| P2S     | 1.482 | **0.878** | _1.435_   | 2.604 | 1.773  | 1.586 | 1.544           |
| Normal  | 2.807 | _2.652_  | 2.837      | 3.265 | **2.648** | 3.723 | 3.628        |

## 4.2 The real-time pipeline

To perform real-time monocular human reconstruction, we design a three-stage pipeline. In the first stage, we get an image from the video stream with OpenCV [4], and use RVM [16] to get a human mask to remove the background. RVM is originally a matting method, and we use the threshold of 0.5 to obtain a binary mask. In the second stage, the FOF of the human is estimated by our network in Sec. 3.2. Then the FOF is resized to a proper resolution ($256 \times 256$ in our implementation), and transformed to a mesh as introduced in Sec. 3.3. In the final stage, we render the mesh into images with PyTorch3D [21] to preview the reconstructed geometry in real time. Our three stages are all implemented with PyTorch and running on a single RTX-3090 GPU. Our pipeline works at 30FPS which is limited by the camera. With a high-frame-rate camera, TensorRT and more GPUs, our frame rate can be further improved. Fig. 4 gives some real-time reconstruction results. More results are available at *http://cic.tju.edu.cn/faculty/likun/projects/FOF*.

## 4.3 Comparison

We compare our method with three state-of-art approaches, PIFu [22], PIFuHD [23] and ICON [27], together with our two variants, FOF-Normal and FOF-SMPL. We retrain PIFu on our dataset. Other methods are used with the pre-trained checkpoints, because they do not provide the training codes. We use $512 \times 512 \times 512$ resolution for all these methods. We do not compare our method with [3] and [10] because they do not release their codes. For simplicity, we also ignore comparisons with some works that have already been compared like PaMIR [30]. We compare our method with two versions of ICON. In Table 2, ICON is the original version proposed in their paper, and ICON$_{filter}$ is provided in their code, which is visually better but may have slightly worse metrics.

**Quantitative results.** Table 2 gives Chamfer distances, P2S distances, and normal image errors on the test set. Ground-truth SMPL meshes are used in FOF-SMPL and ICON [27]. Because of the weak perspective camera, the results of all the methods may not be aligned with the ground-truths on the view direction. Thus, we randomly sample 400000 points on the estimated mesh and the ground-truth mesh, and align them by making their $z$-coordinates have the same mean. Our results have the lowest reconstruction errors compared with others, which are slightly worse than PIFuHD in terms of the normal image error, coming in the second place.

**Qualitative results.** Fig. 5 and Fig. 6 show some visual results for challenging poses and various shapes, respectively. As shown in the figures, our method is robust to various poses and shapes, and achieves detailed and reasonable reconstruction. FOF uses bandlimited approximation (keeping only the first 31 frequencies in our implementation), inevitably discarding high-frequency components of detailed geometries. However, this also makes FOF capable of avoiding high-frequency artifacts

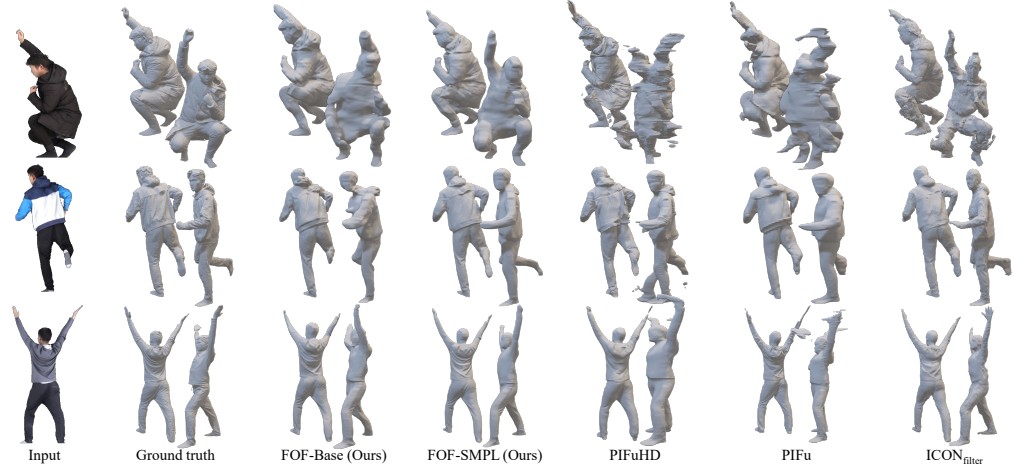

| Input | Ground truth | FOF-Base (Ours) | FOF-SMPL (Ours) | PIFuHD | PIFu | ICON_filter |

Figure 5: Comparison with other methods from a single image with challenging human poses.

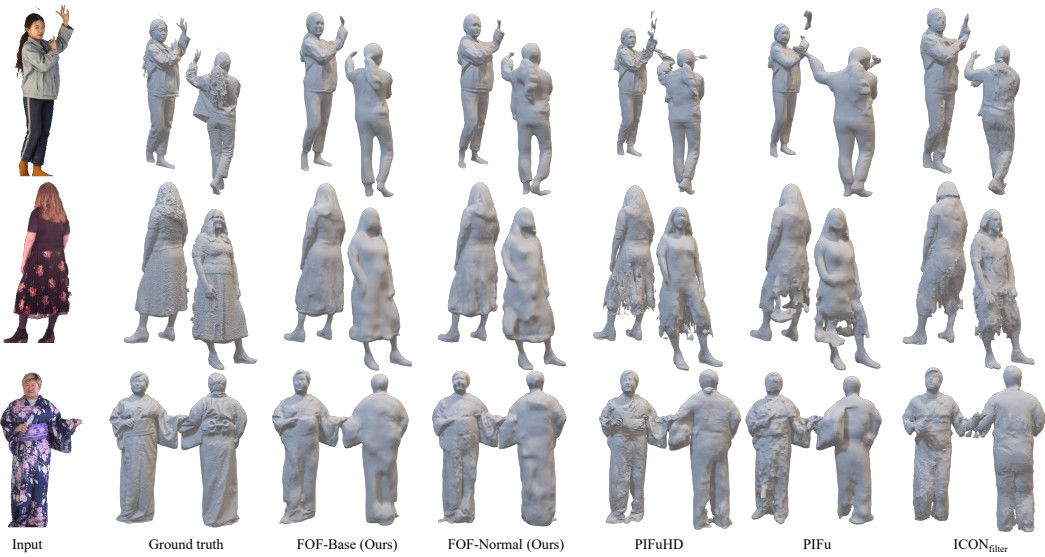

| Input | Ground truth | FOF-Base (Ours) | FOF-Normal (Ours) | PIFuHD | PIFu | ICON_filter |

Figure 6: Comparison with other methods from a single image with various clothes.

as shown Fig. 5 and Fig. 6. This is also consistent with the quantitative results in Sec. 4.3. PIFuHD uses a network with larger capacity in a coarse-to-fine pipeline and visually better, which is more computation and memory demanding. These strategies can also be used for our FOF-based reconstruction to enhance the results but would inevitably make the pipeline more cumbersome to some extent and might sacrify the merit of real-time implementation.

### 4.4 Ablation study

We show the effect of different $N$ on the accuracy of the approximation in Fig. 7. When $N \leq 12$, obvious artifacts appear on the meshes. While the setting of $N = 15$ preserves most geometric details, which is used in our implement. More details can be found in the supplementary document.

## 5 Conclusion and discussion

**Conclusion.** We propose a novel efficient and flexible representation, Fourier Occupancy Field (FOF), for monocular real-time human reconstruction. FOF is aligned with the input image and

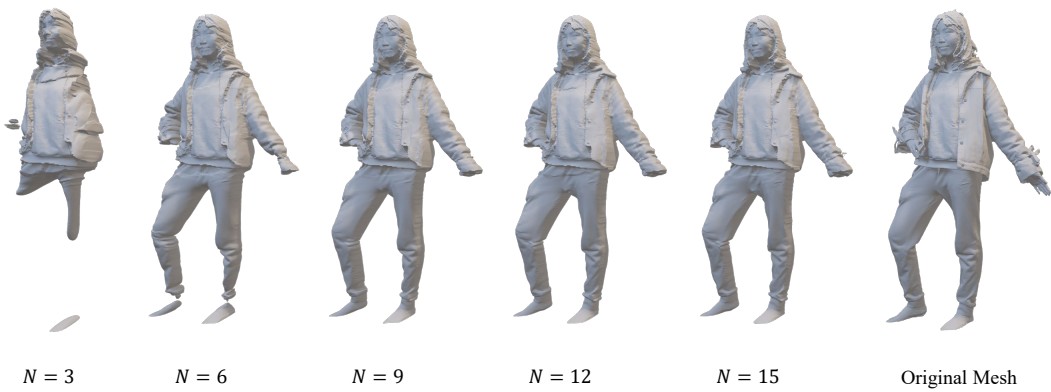

| $N = 3$ | $N = 6$ | $N = 9$ | $N = 12$ | $N = 15$ | Original Mesh |

Figure 7: FOF with different $N$

can be estimated with a simple image-to-image network. Moreover, FOF can represent a complex 3D shape in the form of a multi-channel image, while preserving most geometric details, which is computationally efficient to extract the mesh and suitable for 3DTV applications. Experimental results indicate that our method can infer high-fidelity human models in real-time with state-of-the-art reconstruction quality.

**Limitations.** The FOF representation is based on the Fourier series expansion of the square-wave-like function. When the duty cycle of the square-wave function is small, the spectrum of the function will contain many high-frequency components. In such cases, we need more terms in the Fourier series to approximate the function. The number of terms we need is roughly inversely proportional to the duty cycle. Therefore, FOF cannot represent the objects that are too thin. In future work, we will try wavelet transformation for better representation capability and apply our representation to common objects. Moreover, our FOF representation has great potential for other related tasks, such as 3D shape generation, completion of 3D human bodies and monocular 3D reconstruction for other shapes.

**Broader Impact.** The proposed method will promote the development of avatar generation and 3DTV, which is useful for applications that require accurate and efficient 3D reconstruction. However, this efficient acquisition may cause privacy and ethical problems. We suggest policymakers to establish an efficient regulatory system and inform users about potential risks to avoid the disclosure of personal information.

## 6 Acknowledgements

This work was supported in part by the National Natural Science Foundation of China (62122058, 62171317 and 62231018).

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
