# Supplementary Document for
# FOF: Learning Fourier Occupancy Field for Monocular Real-time Human Reconstruction

**Qiao Feng**
Tianjin University
fengqiao@tju.edu.cn

**Yebin Liu**
Tsinghua University
liuyebin@mail.tsinghua.edu.cn

**Yu-Kun Lai**
Cardiff University
laiy4@cardiff.ac.uk

**Jingyu Yang**
Tianjin University
yjy@tju.edu.cn

**Kun Li**[*]
Tianjin University
lik@tju.edu.cn

In this document, we provide more paper details, including:

- Training details;
- Mesh to FOF algorithm;
- About our project;
- About our data;
- Ablation study on the *N*;
- Running time of each component;
- Visualization of 1D occupancy curves;
- Relative training loss;
- Results on manually added noise.

## 1 Training details

Since our FOF is not limited to a specific network architecture, any image-to-image network can be used as our backbone. HR-Net-W32-V2 [2] is used in our implementation with a simple decoder head. The surface normal used in FOF-Normal is inferred with the same networks in PIFuHD [1]. Notice that we use the inferred normal maps, instead of ground-truth normal maps, in the training stage. Our method is trained on an RTX 3090 GPU for 10 epochs using the Adam optimizer with the learning rate of 0.0002 and the batchsize of 8.

## 2 Mesh to FOF algorithm

To transform a 3D mesh to a FOF, we first extract the discontinuous points $\mathcal{Z} \triangleq \{(z_i, z_i')\}_{i=1}^{k}$ for each pixel on the FOF with a rasterizer-like procedure, and then get the FOF with Eq. (6) in the main paper. Here we describe the rasterizer-like algorithm in details. Denote by $n$ and $m$ the number of vertices and faces of the mesh, respectively. Given vertices $\{v_i\}_{i=0}^{n}$ and faces $\{f_j\}_{j=0}^{m}$ of the mesh, we solve for the covered pixels for each $f_j$ and add the corresponding depth values $z$ to the set of the pixels $\mathcal{Z}$. Then, we sort the set of each pixel and pair the depth values in them to get the final result. This is the simplest implementation of our rasterizer-like algorithm, and it can be improved with other rasterizer algorithms such as line sweep.

36th Conference on Neural Information Processing Systems (NeurIPS 2022).

## 3  About our project

Our real-time reconstruction project contains the following third-party projects:

- RVM[2] for video segmentation, which is under the GNU General Public License v3.0.
- Torchmcubes[3] for iso-surface extraction, which is under the MIT License.

With TensorRT, our real-time pipeline can transform an input image to an occupancy voxel grid at the speed of 70FPS, and to a mesh at the speed of 38FPS. Because we use the most common USB webcam (Logitech C920 PRO) in our real-time system, which can only run at 30FPS, the total speed of our system is limited. In fact, our system could be further improved. The background matting stage is not necessary, and the marching cubes algorithm can also be accelerated with the octree. Our code will be released after the paper is accepted.

## 4  About our data

We use Twindom[4] and THuman2.0 [3] datasets in our paper. The signed contract with the people in the datasets is the ownership and perpetual use of the data.

## 5  Ablation study on the $N$

Table 1: Quantitative ablation study results on the number of FOF components.

| $N$ | 3 | 7 | 15 | 31 | 63 | 127 |
|---|---|---|---|---|---|---|
| P2S Distance | 1.473 | 0.535 | 0.179 | 0.075 | 0.045 | 0.038 |
| Chamfer Distance | 6.926 | 0.939 | 0.354 | 0.203 | 0.150 | 0.141 |

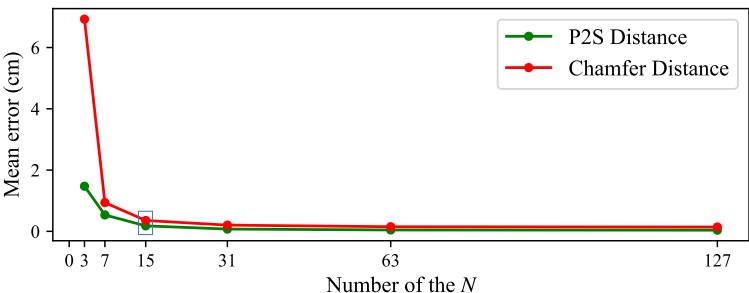

Figure 1: Quantitative ablation study curves on the number of FOF components.

We calculate the ground-truth FOF with different number of components and measure the geometric errors of the reconstructed meshes, which is shown in Table 1 and Fig. 1. It can be seen that, the approximations are good enough with $N = 15$. Some small improvements can be made with $N = 31$, but it makes no sense to choose a larger $N$.

## 6  Running time of each component

The running time of each component is shown in Table 2. Note that, the three stages are in parallel to achieve a real-time pipeline, and the most time-consuming stage takes about 20ms to process the data. The pipeline cannot run in full speed due to the limitation of the 30FPS camera. The neural network and the tensor multiplication are integrated into one component implemented with TensorRT for further acceleration.

---

[2]https://github.com/PeterL1n/RobustVideoMatting
[3]https://github.com/tatsy/torchmcubes
[4]https://web.twindom.com/

Table 2: time break down.

| Component | Stage 1 | | Stage 2 | | Stage 3 |
|---|---|---|---|---|---|
| | Streamer | Segmentation | Inference (Ours) | Marching Cubes | Rendering |
| Time (ms) | 8.21 | 8.33 | 1.86 | 16.18 | 20.53 |
| Implementation | OpenCV | PyTorch | TensorRT | Pytorch Extension | PyTorch3D |

# 7    Visualization of 1D occupancy curves

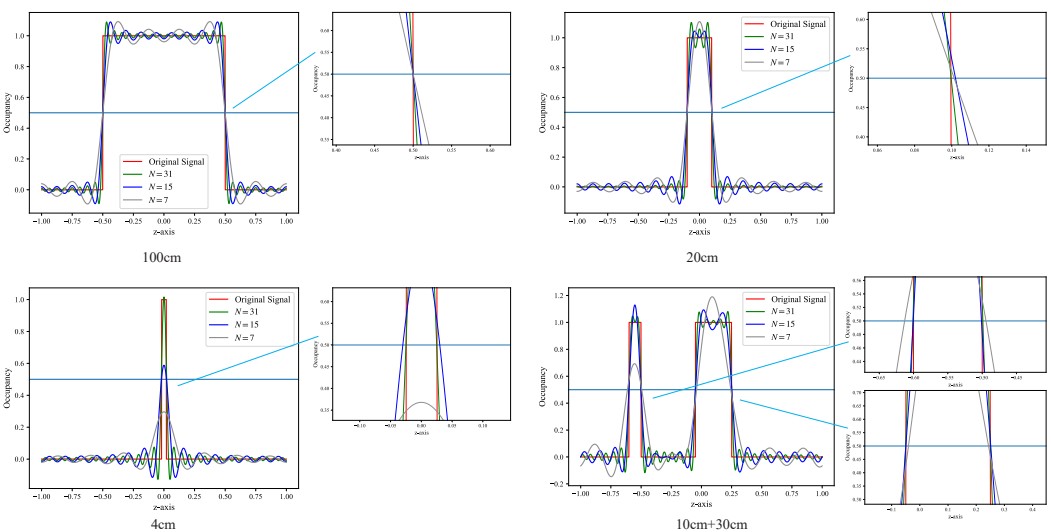

Figure 2: Visualization of 1D occupancy curves.

Fig. 2 shows the approximations with different number of Fourier basis functions for some typical signals. Note that, the translation of the signal in space domain does not affect the accuracy of the approximation. Thus, only the case of different signal widths is given here. We should focus on the intersection of the curve and the horizontal line, which is the position of the surface. As can be seen, all three approximations ($N = 7, N = 15, N = 31$) work well in general (the two sub-figures in the first line). But the approximation with fewer terms ($N = 7$, the gray curve) performs poorly with multiple layers (bottom right) and cannot represent thin objects (bottom left).

# 8    Relative training loss

Table 3: Relative training loss.

| Epoch | 1 | 2 | 3 | 4 | 5 | 6 | 7 | 8 | 9 | 10 |
|---|---|---|---|---|---|---|---|---|---|---|
| Mean Error | 38.7% | 30.1% | 26.1% | 23.8% | 22.2% | 21.0% | 20.3% | 19.5% | 18.9% | 18.6% |
| Low Frequency Error | 26.5% | 20.0% | 17.3% | 16.0% | 15.0% | 14.2% | 13.8% | 13.4% | 13.0% | 12.8% |
| High Frequency Error | 50.1% | 39.5% | 34.4% | 31.2% | 29.0% | 27.4% | 26.4% | 25.3% | 24.4% | 23.9% |

The relative training loss is shown in Table 3 and Fig. 3. It can be found that the high-frequency error and the low-frequency error converge simultaneously, and the low-frequency error is much smaller than the high-frequency error which makes the results more robust.

# 9    Results manually added noise

We manually add relative Gaussian noise with different variances and shows the quantitative results in Fig. 4 and Table 4. Fig. 5 shows some qualitative results. Based on these experiments, we can conclude that the FOF is very robust and can keep the shape even with large noise.

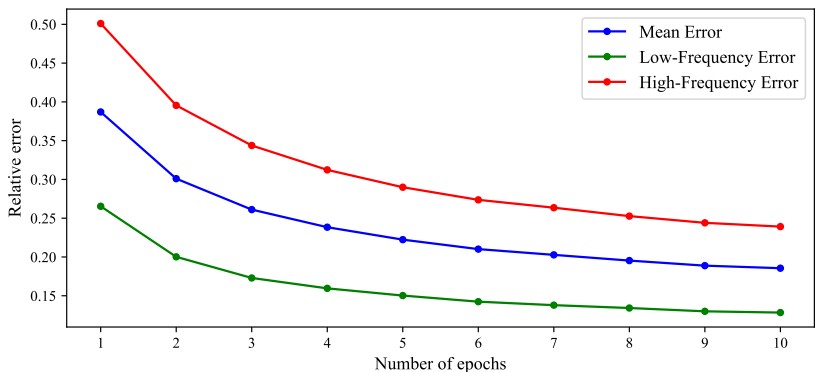

Figure 3: Relative training loss.

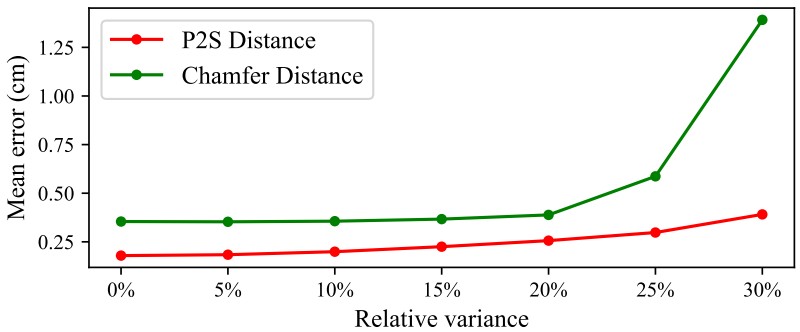

Figure 4: Quantitative results with noise on FOF.

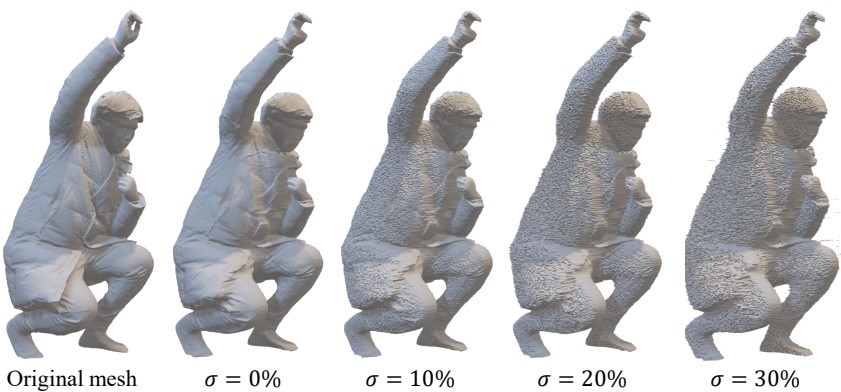

Original mesh    $\sigma = 0\%$    $\sigma = 10\%$    $\sigma = 20\%$    $\sigma = 30\%$

Figure 5: Visual results with noise on FOF.

Table 4: Quantitative results with noise on FOF.

| Relative Variance | 0% | 5% | 10% | 15% | 20% | 25% | 30% |
|---|---|---|---|---|---|---|---|
| P2S Distance | 0.179 | 0.184 | 0.199 | 0.225 | 0.256 | 0.297 | 0.391 |
| Chamfer Distance | 0.354 | 0.354 | 0.356 | 0.367 | 0.388 | 0.587 | 1.391 |