# OpenReview forum: "FOF: Learning Fourier Occupancy Field for Monocular Real-time Human Reconstruction"
_NeurIPS.cc/2022/Conference — NeurIPS 2022 Accept_

### Official Review · Reviewer_mkLu · 2022-07-08

**Rating:** 7
**Confidence:** 4
**Soundness:** 3 good
**Presentation:** 3 good
**Contribution:** 3 good

**Summary:**

This paper proposes a new representation for human reconstruction. Specifically, for a pixel in the input image, it approximates the occupancy along the casted ray as the Fourier series. The proposed approach learns the representation from data by regressing to the coefficients of the Fourier basis functions. The submission shows the efficacy and flexibilities of the representation on real captured data.

**Questions:**

Essentially, the training of FOF will be a per-pixel regression problem. Namely, the network needs to regress to GT coefficients $C(x, y)$ for Fourier basis functions. Since the per-pixel regression is a hard problem, I am curious about how robust the FOF representation is to the inaccuracy of such regression. Specifically, I think showing the following results would be beneficial:
1. The training loss of relative error (absolute error may not be that informative as the magnitude matters)
2. Manually add noises to the GT coefficients and plot a curve to show the correlation between the perturbed coefficients and the quantitative quality of the reconstructed meshes.

**Limitations:**

I appreciate the authors's discussions about limitations of the proposed approach in Sec. 5, which helps the understanding of the suitable scenarios.

**Strengths And Weaknesses:**

## Strengths

1. Originality: the proposed representation of FOF is novel. Approximating the occupancy with Fourier series is interesting and principled.
2. Quality: the presented reconstruction is of high-quality.
3. Clarity: the paper is well-written and easy to follow.
4. Significance: it is an important problem of developing an efficient and powerful 3D representation to enable real-time application. The submission demonstrates a real-time pipeline with the FOF representation.

## Weakness

I do not see some major weakness. But it would be great if authors can discuss more about the quality of the reconstruction. For example, in Fig. 4, Fig. 5, and video in the supplementary, I observe that PIFuHD usually maintains more details even though FOF-Normal already utilizes the inferred normal maps. It would be beneficial to discuss why FOF representations are not detailed than PIFuHD.

---

> ### Author Response · Authors · 2022-08-02
> **Response to Reviewer mkLu**
>
> Thanks for your acknowledgement for our work and the constructive comments. We will address your questions and concerns in the following:
>
> > **Q1:** I do not see some major weakness. But it would be great if authors can discuss more about the quality of the reconstruction. For example, in Fig. 4, Fig. 5, and video in the supplementary, I observe that PIFuHD usually maintains more details even though FOF-Normal already utilizes the inferred normal maps. It would be beneficial to discuss why FOF representations are not detailed than PIFuHD.
>
> **Reply:** Regarding the performance in terms of detail recovery, there are two main reasons:
> 1. FOF uses bandlimited approximation (keeping only the first 31 frequencies in our implementation), inevitably discarding high-frequency components of detailed geometries. However, this also makes FOF capable of avoiding high-frequency artifacts as shown Fig. 4 and Fig. 5 of the paper. This is also consistent with the quantitative results in Table 2 of the paper: FOF-based methods have the lowest geometric errors (in Chamfer and P2S) and are marginally worse than PIFuHD in terms of the normal image error.
> 2. PIFuHD uses a network with larger capacity in a coarse-to-fine pipeline, which is more computation and memory demanding. These strategies can also be used for our FOF-based reconstruction to enhance the results but would inevitably make the pipeline more cumbersome to some extent and might sacrify the merit of real-time implementation. The above discussion will be added to the final version.
>
> > **Q2:** The training loss of relative error (absolute error may not be that informative as the magnitude matters)
>
> **Reply:** Thanks for your suggestion. The training loss of relative error has been added in Sec. 8 of the supplementary document.
>
> > **Q3:** Manually add noises to the GT coefficients and plot a curve to show the correlation between the perturbed coefficients and the quantitative quality of the reconstructed meshes.
>
> **Reply:** We have manually added noises to the GT coefficients and shown the results in Sec. 9 of the supplementary document.

---

> > ### Comment · Reviewer_mkLu · 2022-08-09
> > **Thanks for addressing my concerns**
> >
> > I thank authors' time and effort to answer my questions. I think the newly-added results are informative and demonstrate FOF's robustness. I maintain my score for this work.

---

### Official Review · Reviewer_AEcD · 2022-07-12

**Rating:** 5
**Confidence:** 4
**Soundness:** 3 good
**Presentation:** 3 good
**Contribution:** 2 fair

**Summary:**

This paper proposed an occupancy field representation for monocular real-time human reconstruction. The proposed representation represents a 3D object with a 2D field orthogonal to the view direction. Each 2D position stores the 1D occupancy field of the object along the view direction, and the 1D occupancy field is represented by real-valued fourier series. The proposed method is able to achieve 30+FPS while achieving comparable accuracy to other monocular methods.

**Questions:**

(1) With the fourier transformation of 1D the occupancy field given in Equation 6, what is the error bounds of the distance between recovered v.s. groundtruth surface point?

(2) More visualization of the recovered 1D occupancy curves.

(3) Discussion of different ways to represent bandlimited signals and its implication in the problem of interest.

(4) In the discussion, the authors mentioned future works could incorporate wavelet transforms to improve results for thin objects. But it seems that would blow up the dimension of the representation with an naive implementation, and its computational advantage over voxel-based representation may diminish. I would be curious to learn if the idea of utilizing basic signal processing techniques advocated by the authors could indeed be generalized to more complex scenarios.



**Ethics Review Area:**

["I don’t know"]

**Limitations:**

As pointed out by the author, the bandlimited approximation of the 1D occupancy field is insufficient to recover fine-details as well as thin structures. The poor results in hand and limb regions reveal such defects.

**Strengths And Weaknesses:**

Strength: The paper proposes a real time monocular human reconstruction method with implicit occupancy representation. The occupancy field is represented as a pixel aligned 2D grid. The novel idea of this paper is, in each grid the 1D occupancy signal is represented by real-valued fourier series.

The paper shows that 1D occupancy signals tend to be low frequency, thus can be effectively represented by the first few fourier basis functions. This reduces the output dimension of the inference model, thus effectively improves computational efficiency.

Weakness: approximating occupancy with a very small number of fourier basis is still quite counter intuitive. I think the author could do a better job in analyzing the property of this approximation. It would be nice to see what is the theoretical bound of the distance to the surface given Equation 6. It would also be nice if the authors visualize the reconstructed 1D occupancy curve given different number of fourier basis. In addition, representing signals with a discrete set of uniformly sampled fourier basis is equivalent to applying sinc filter to regular-grid delta signals. In other words, there are different ways of representing a band-limited signal (e.g. spatial vs frequency representation), thus it would be helpful if the author provides some insight what is the benefits of representing signals in the frequency v.s. spatial domain.

In general, I recognize the practical advantage of the proposed method, but feels the technical / theoritical contribution of the paper is a bit thin. The paper in general follows the engineering details of other methods such as PIFU and ICON and the difference is mainly the simplified way of representing occupancy signals with fourier transforms. Such strategy has been seen in use in other problems, and it inhirits the defects of bandlimited approximations.

---

> ### Author Response · Authors · 2022-08-02
> **Response to Reviewer AEcD (2/2)**
>
> > **Q4:** Discussion of different ways to represent bandlimited signals and its implication in the problem of interest.
>
> **Reply:** Besides Fourier representation, there are other possible ways to represent the occupancy signals, which are not bandlimited signals but can be approximated with bandlimited signals:
>
> 1. As the first work to represent a 3D geometry with a 2D field in frequency domain, we enable memory efficiency, real-time implementation, high quality, robustness, and generality for monocular human reconstruction. Instead of simply adopting the discrete Fourier transform, we devise the FOF representation with Fourier series, achieving sampling scalability. With this merit, our method can adapt to different sampling rates in the inference stage without introducing systematic sampling mismatch. The benefits of analysis in frequency domain have been verified in in various fields [R1], [R2], [R3].
> 2. For fast inference, we choose the Fourier series representation, and use linear approximation of the first (2N+1) terms in our scheme for its promising sampling scalability and fast implementation. The loss of high-frequency information results in a lack of some visual details, but also makes the FOF more robust without high-frequency artifacts, which is confirmed by Table 2 in the paper.
> 3. Wavelet transforms with better space(time)-frequency localization are potentially more efficient in representing signals with singularities by nonlinear approximation (picking the M largest terms). However, the linear approximation (using the first M terms) of wavelet transform has the same order of approximation error as the Fourier presentation [R4] and can blow up the dimension of the representation as mentioned in your Q5.
>
> > **Q5:** In the discussion, the authors mentioned future works could incorporate wavelet transforms to improve results for thin objects. But it seems that would blow up the dimension of the representation with an naive implementation, and its computational advantage over voxel-based representation may diminish. I would be curious to learn if the idea of utilizing basic signal processing techniques advocated by the authors could indeed be generalized to more complex scenarios.
>
> **Reply:** As discussed in the response to Q4, a straightforward application of a discrete wavelet transform would diminish several advantages of our method such as fast implementation and sampling scalability. Instead, it is possible to borrow the multiscale idea of wavelet to build a hierarchical inference structure of space-frequency representation. It is worthy of exploring nonlinear approximation to lower the reconstruction error bound while keeping fast implementation. It is also important to devise a particular family of multiscale representation to achieve sampling scalability similar to Fourier series representation.
>
> Moreover, besides human bodies, our FOF-based reconstruction can be also applied to more complex scenes. We will show some examples on the project page when the code is released and mention more potential applications in final version.

---

> ### Author Response · Authors · 2022-08-02
> **Response to Reviewer AEcD (1/2)**
>
> Thanks for your acknowledgement for our work and the constructive comments. We will address your questions and concerns in the following:
>
> > **Q1:** The paper in general follows the engineering details of other methods such as PIFU and ICON and the difference is mainly the simplified way of representing occupancy signals with fourier transforms. Such strategy has been seen in use in other problems, and it inhirits the defects of bandlimited approximations.
>
> **Reply:** Thanks for your careful review. Our work is not a trivial combination of Fourier transform and engineering details of other methods:
> + As the first work to represent a 3D geometry with a 2D field in frequency domain, we enable memory efficiency, real-time implementation, high quality, robustness, and generality for monocular human reconstruction. Instead of simply adopting the discrete Fourier transform, we devise the FOF representation with Fourier series, achieving sampling scalability. With this merit, our method can adapt to different sampling rates in the inference stage without introducing systematic sampling mismatch. The benefits of analysis in frequency domain have been verified in in various fields [R1], [R2], [R3].
> + Compared with existing methods, we provide a compact and simple framework with an image-to-image network benefitting from our FOF representation, which achieves real-time, high-quality and robust reconstruction without any engineering tricks.
> + Our FOF has better compatibility and extensibility. In the original paper, we showed two variants, FOF-SMPL and FOF-Normal, which use SMPL as a prior or use normal maps to enhance the visual results. Please note that we present a new way to exploit the SMPL as a prior, which is completely different from PaMIR and ICON.
>
> [R1] J. Lin, Y. Liu, J. Suo and Q. Dai, "Frequency-Domain Transient Imaging," in IEEE Transactions on Pattern Analysis and Machine Intelligence, vol. 39, no. 5, pp. 937-950, 1 May 2017, doi: 10.1109/TPAMI.2016.2560814.
>
> [R2] Wang, L. , et al. "Fourier PlenOctrees for Dynamic Radiance Field Rendering in Real-time." in Proceedings of CVPR2022.
>
> [R3] Changjian Zhu, Hong Zhang, Qiuming Liu, Yanping Yu, and Hongtao Su, "Frequency analysis of light field sampling for texture information," Opt. Express 28, 11548-11572 (2020)
>
> > **Q2:** With the fourier transformation of 1D the occupancy field given in Equation 6, what is the error bounds of the distance between recovered v.s. groundtruth surface point?
>
> **Reply:** Following the analysis in [R4], we measure the approximation error of occupancy signals by the expected squared error, which is formulated as: $\epsilon=E\left[ \left\Vert \hat{f}(z)-f(z) \right\Vert_2^2 \right] = E\left[ \int_{-1}^{1} \left| \hat{f}(z)-f(z) \right|^2 dz\right]$, where $f(z), z\in [-1, 1]$ is the original occupancy signal, and $\hat{f}(z)$ is the approximated signal. Note that occupancy signals belong to the family of piecewise constant functions that are constant expect for a few jumps (discontinuity points). For the presence of discontinuity, the Fourier series coefficients decay by the following rate:$\left| a_n \right|, \left| b_n \right| \sim 1/n$, where $a_n$ and $b_n$ are the cosine and sine coefficients at frequency $2\pi n$ as defined in the paper. The linear approximation error by the first $2N+1$ terms (the same as in the paper) has the following order: $\epsilon_N \sim \sum_{n=2N+1}^{\infty}1/n^2 \sim 1/N$, where $\epsilon_N$ is the expected squared error with the first $2N+1$ terms for approximation [R4]. In summary, the approximation error bound of occupancy signals in FOF representation has the order of $1/N$, and the approximation would be more accurate as the increasing of the number of retained terms. The selection of N is also discussed in the reply of Q3 from Reviewer mKrW.
>
> However, it is difficult to analyze the theoretical bound of overall reconstruction error since our overall reconstruction pipeline does not only include the Fourier approximation, but also involves isosurface extraction using Marching Cubes, which means minor differences that do not cross the inside/outside threshold will have no impact on the reconstructed shape. This nonlinear operation also impedes strict mathematical analysis of the overall method. But, we reach a conclusion in a loose sense: the overall surface reconstruction error is lower than the bound above ($\epsilon_N \sim 1/N$) as the surface fitting in Marching Cubes serves as a kind of filter to further reduce the 3D geometrical errors.
>
> [R4] M. Vetterli, "Wavelets, approximation, and compression," in IEEE Signal Processing Magazine, vol. 18, no. 5, pp. 59-73, Sept. 2001, doi: 10.1109/79.952805.
>
> > **Q3:** More visualization of the recovered 1D occupancy curves. It would also be nice if the authors visualize the reconstructed 1D occupancy curve given different number of fourier basis.
>
> **Reply:** We have visualized the curves in Sec. 7 of the supplementary document.

---

### Official Review · Reviewer_mKrW · 2022-07-13

**Rating:** 5
**Confidence:** 5
**Soundness:** 3 good
**Presentation:** 3 good
**Contribution:** 3 good

**Summary:**

The paper proposes a Fourier Occupancy Field representation to address the problem of monocular human mesh reconstruction from an image. In FOF, the 3D mesh is essentially converted into a multi-channel image that also possesses sampling scalability, stable reconstruction, and low-complexity reconstruction. Experiments demonstrate the proposed method outperforms existing methods quantitatively and qualitatively, while also enabling real-time applications.

**Questions:**

- Regarding the number of FOF components: It would be great to have a quantitative analysis of how this affects the performance. Also, it seems this number doesn't significantly impact run time since only tensor multiplication is needed. It would be great if the authors could explain why a maximum of 15 is used, i.e., what would happen if N>100?
- Would the FOF representation be used in other related tasks such as general single image 3D prediction?

**Limitations:**

The authors have adequately addressed the limitations and potential negative societal impact of their work.

**Strengths And Weaknesses:**

The strengths of the paper are:
- The paper is well-written. The discussions and explanations are not only informative but also insightful.
- The proposed FOF representation is simple and effective. The methodology is general purpose and could benefit other related tasks.

The weaknesses of the paper are:
- Minor improvements on paper writing: 1) A typo in Line 236 on the symbol subscript. 2) Table 1 would be much more informative if references could be added for each type of method. 3) It would be nice to specify if any post-processing techniques are used such as smoothing or connected components.
- There is a significant performance gap, both quantitatively and qualitatively, between vanilla FOF and the enhanced version FOF-SMPL. However, in the main methodology section, the SMPL part was somehow downplayed. If this was done to emphasize FOF as the main novelty, it would be more convincing to apply FOF to other relevant tasks. Otherwise, if the focus is on 3D human reconstruction, the SMPL part should be included as an important subsection with more details.
- The experimental section would be further richened. For instance, a quantitative ablation study on the number of FOF components would be great to help understand how it works. A runtime break-down would be also nice to shed light on how fast each component is.

---

> ### Author Response · Authors · 2022-08-02
> **Response to Reviewer mKrW**
>
> Thanks for your acknowledgement for our work and the constructive comments. We will address your questions and concerns in the following:
>
> > **Q1:** Minor improvements on paper writing: 1) A typo in Line 236 on the symbol subscript. 2) Table 1 would be much more informative if references could be added for each type of method. 3) It would be nice to specify if any post-processing techniques are used such as smoothing or connected components.
>
> **Reply:** Thanks for your careful review. We have revised the paper following your suggestions. We do not use any post-processing techniques in our implementation. All our results are produced directly by the Marching Cubes algorithm from the inferred FOF representation. This has been emphasized in the revised version.
>
> > **Q2:** There is a significant performance gap, both quantitatively and qualitatively, between vanilla FOF and the enhanced version FOF-SMPL. However, in the main methodology section, the SMPL part was somehow downplayed. If this was done to emphasize FOF as the main novelty, it would be more convincing to apply FOF to other relevant tasks. Otherwise, if the focus is on 3D human reconstruction, the SMPL part should be included as an important subsection with more details.
>
> **Reply:** Thanks for your suggestion. This work focuses on 3D human reconstruction, and we will include the SMPL part as an important subsection in the final version due to the limitation of 9 pages at this stage.
>
> > **Q3:** The experimental section would be further richened. For instance, a quantitative ablation study on the number of FOF components would be great to help understand how it works. A runtime break-down would be also nice to shed light on how fast each component is.
>
> **Reply:** Additional experiments as requested have been added in Sec. 5 and Sec. 6 of the supplementary document.
>
> > **Q4:** Regarding the number of FOF components: It would be great to have a quantitative analysis of how this affects the performance. Also, it seems this number doesn't significantly impact run time since only tensor multiplication is needed. It would be great if the authors could explain why a maximum of 15 is used, i.e., what would happen if N>100?
>
> **Reply:** Using a larger number of FOF components (N) does not affect the run time of the reconstruction part that involves only the multiplication of the basis tensor and the coefficient tensor. However, the number of channels in previous convolutional layers should be also enlarged to increase the network capacity otherwise the performance will actually drop. As a result, increasing N will also enlarges the overall network, and requires more computational and memory resources of GPU. Moreover, the training of the network will also become more difficult. As shown in ablation results suggested by Q3, the reconstruction performance reaches the knee point at N=15, and a larger N beyond will bring only marginal improvements.
>
> > **Q5:** Would the FOF representation be used in other related tasks such as general single image 3D prediction?
>
> **Reply:** Yes, our FOF representation can be used for many tasks, such as 3D shapes generation, completion of 3D human bodies and monocular 3D prediction for other shapes. We will show some examples on the project page when the code is released, and mention these potential applications in final version.

---

### Author Response · Authors · 2022-08-09
**Have our replies addressed your concerns?**

Dear Reviewers:

Thank you very much for your time and effort in reviewing our paper. It is less than **15 hours** before the end of the Author-Reviewer Discussion and we have not received any response yet. We wonder if our replies address your concerns. Our updated supplementary document includes new experimental results you are concerned with. Thank you very much.

Best regards!

---

### Meta-Review · Area_Chair_i5wk · 2022-08-28

**Recommendation:** Accept
**Confidence:** Certain

**Metareview:**

This paper received 3 positive reviews: 2xBA + A. All reviewers acknowledged that this work introduces meaningful and non-trivial contributions, it is well presented, and the claims are supported by strong empirical performance. The remaining questions and concerns were addressed in the authors' responses, which seemed convincing to the reviewers.
The final recommendation is therefore to accept.

**Award:**

No

---

### Decision · Program_Chairs · 2022-09-14

Accept